# Ethyl Acetate Fractions of Tectona Grandis Crude Extract Modulate Glucose Absorption and Uptake as Well as Antihyperglycemic Potential in Fructose–Streptozotocin-Induced Diabetic Rats

**DOI:** 10.3390/ijms25010028

**Published:** 2023-12-19

**Authors:** Olakunle Sanni, Pilani Nkomozepi, Md. Shahidul Islam

**Affiliations:** 1Department of Human Anatomy and Physiology, University of Johannesburg, Doornfontein, Johannesburg 2028, South Africa; kunlesola2810@gmail.com (O.S.); pilanin@uj.ac.za (P.N.); 2Department of Biochemistry, School of Life Sciences, University of Kwazulu-Natal (Westville Campus), Private Bag X54001, Durban 4000, South Africa

**Keywords:** pancreatic β cell, hyperglycemia, glucose tolerance, *Tectona grandis*

## Abstract

Type 2 diabetes (T2D) is a global health challenge with increased morbidity and mortality rates yearly. Herbal medicine has provided an alternative approach to treating T2D with limited access to formal healthcare. *Tectona grandis* is being used traditionally in the treatment of diabetes. The present study investigated the antidiabetic potential of *T. grandis* leaves in different solvent extractions, and the crude extract that demonstrated the best activity was further fractionated through solvent–solvent partitioning. The ethyl acetate fraction of the ethanol crude extract showed the best antidiabetic activity in inhibiting α-glucosidase, delaying glucose absorption at the small intestine’s lumen, and enhancing the muscle’s postprandial glucose uptake. The ethyl acetate fraction was further elucidated for its ability to reduce hyperglycemia in diabetic rats. The ethyl acetate fraction significantly reduced high blood glucose levels in diabetic rats with concomitant modulation in stimulated insulin secretions through improved pancreatic β-cell function, insulin sensitivity by increasing liver glycogen content, and reduced elevated levels of liver glucose-6-phosphatase activity. These activities could be attributed to the phytochemical constituents of the plant.

## 1. Introduction

Diabetes mellitus has posed a global health challenge over the last few decades with the unrelenting increase in the number of people living with diabetes. Data from the International Diabetes Federation (IDF) revealed that type 2 diabetes (T2D) accounted for over 90% of diabetic cases globally. T2D is characterized by prolonged hyperglycemia due to dysfunction in insulin secretion and cells’ failure to respond to insulin action [1]. Persistent hyperglycemia has been implicated in the pathogenesis of diabetic complications, such as retinopathy, neuropathy, foot ulcers, cardiovascular disease [2], etc.

The pathogenesis of T2D is well documented and involves the failure of cells to respond to the action of insulin, which increases the hyperfunction of β ells to produce more insulin, consequently developing into β-cell exhaustion and dysfunction [3,4]. Hence, therapeutic interventions have been developed to either increase the insulin sensitivity of the cells or improve the functionality of pancreatic β cells. For example, Sulfonylureas (glibenclamide) increase insulin concentrations, while biguanides (metformin) inhibit glucose production in the liver and increase insulin-mediated skeletal muscle glucose uptake [5]. However, most T2D patients combined these intervention classes to optimise the control over their blood glucose levels. As a result, the side effects of these antidiabetic drugs are a staggering burden on patients. Therefore, the effort to find new oral anti-diabetic agents is never ending.

Recently, the use of medicinal plants in the treatment of degenerative diseases has gained tremendous attention due to their availability and little or no side effects [6]. Also, the folkloric use of medicinal plants in diabetic treatment has attracted the attention of researchers for their scientific validation. Among such medicinal plants is *Tectona grandis* (*T. grandis*). *T. grandis*, a teak tree, is an indigenous medicinal plant of sub-Saharan Africa, Asia, and North America. It has a folk reputation for treating diabetes, inflammation, ulcers, lipid disorders, and tuberculosis [7,8].

Previous studies have demonstrated the biological activity of *T. grandis*, such as healing activity [9], antitumor activity [10], antibacterial and antifungal activity [11], antioxidant activity [12], and anti-inflammatory activity [13]. There are reports on the antidiabetic activity of *T. grandis* using different parts of the plant and different solvents of extraction, such as methanol extract from the flowers [14], methanolic extract from the roots [15], and methanolic extract of the stem bark [15,16]. However, most of these studies failed to demonstrate the mechanism of the antidiabetic activity or highlight the bioactive compounds responsible for the activity.

Recently, the antidiabetic activity of *T. grandis* was investigated using in vitro antidiabetic enzymes (α glucosidase) and glucose uptake assay in 3T3-L1 cells using different fractions of the methanolic extract of the leaves. The petroleum ether extract and ethanol fractions showed significant glucose uptake and α-glucosidase inhibitory activity [13]. The study, however, considered some questions on (1) the role of solvents of extraction on their antidiabetic activity, (2) the bioactive compounds in these solvents and their relationship to the antidiabetic activity, and (3) the cumulative effect of the bioactive compound and mechanism of their action.

Hence, we designed our study to compare the antidiabetic activity of different extracts; after that, we fractionated the extracts and determined the fractions with the highest activity. Finally, the fraction that demonstrated the highest activity was investigated in fructose–streptozotocin (STZ)-induced diabetic rats to determine the possible mechanism of action.

## 2. Results

### 2.1. Total Phenol Content of the Crude and the Fractions of Ethanol Extract

The result of the total phenol of the crude extract from *T. granndis* leaves and the ethanol crude extract fractions is presented in Table 1. The ethanol crude extract is significantly (*p* < 0.05) higher than the remaining crude extracts, whereas the total phenol content of the ethyl acetate fraction is significantly (*p* < 0.05) higher than the remaining fractions.

### 2.2. Sequential Extraction of T. grandis Leaves and Antidiabetic Activity

After the sequential extraction of *T. grandis* leaves, crude ethanol extract demonstrated the highest α-glucosidase inhibitory activity compared to the remaining crude extracts (Figure 1). The IC_50_ of different crude extracts (ethyl acetate, ethanol, and aqueous) on α-glucosidase showed that the ethanol extract has the highest inhibitory activity (*p* < 0.05), as compared with other crude extracts (Table 1).

### 2.3. Fractionation of T. grandis Leaves Crude Extract and Antidiabetic Activity Ex Vivo

As a result of 3.1 above, the ethanol crude extract was sequentially partitioned with dichloromethane, ethyl acetate, and butanol, and their respective extracts were obtained and tested for (a) α-glucosidase inhibitory activity, (b) glucose absorption inhibitory activity in the rat intestinal jejunum, and (c) glucose uptake in psoas rat muscle. The incubation of the fractions with α-glucosidase enzyme resulted in a significant (*p* < 0.05) inhibition in a dose-dependent manner, similar to the standard drug, acarbose (Figure 2A). The ethyl acetate showed the best activity, as indicated by the lowest IC_50_ (Table 1). Also, the absorption of glucose by the jejunum upon incubation without the fraction was significantly (*p* < 0.05) higher. But, upon incubation with the fraction, the glucose absorption was significantly (*p* < 0.05) decreased in a dose-dependent manner (Figure 2B), with ethyl acetate fraction demonstrating the highest activity (Table 1). Conversely, the fractions significantly increased the glucose uptake when incubated with psoas muscle in Kreb buffer. The incubation with the fractions increased the glucose uptake in a dose-dependent manner (Figure 2C), with ethyl acetate fractions demonstrating the highest (Table 1).

### 2.4. Ethyl Acetate Fractions Studies in Fructose–Streptozotocin Diabetic Rats

Based on the ex vivo result in Section 2.2, the ethyl acetate fraction of the crude ethanol crude extract showed the highest activity in all the antidiabetic models tested for and consequently selected for further studies in fructose–streptozotocin diabetic rats.

#### 2.4.1. Effect on Animal Body Weight, Food, and Fluid Intake

There was a significant (*p* < 0.05) reduction in the body weight of the diabetic animal upon induction of diabetes, as indicated in Figure 3A. However, during the intervention period, DTGH, DTGL, and DBM rats significantly (*p* < 0.05) and progressively gained weight. The fluid intake was significantly (*p* < 0.05) increased upon the induction of diabetes but the effect on the food intake when compared to the normal rats was not significant. However, treatment with the high dose of ethyl acetate fraction of *T. grandis* and metformin significantly (*p* < 0.05) reduced the fluid intake, while the low dose of the fraction did not reduce the fluid intake significantly (Figure 3B)

#### 2.4.2. Effect on Blood Glucose

The NFBG level of the diabetic groups (DBC, DTGL, DTGH, and DBM) increased significantly (*p* < 0.05) in the induction of diabetes, as compared to normal control groups (NC and NTPI) (Figure 4). However, the treatment of the diabetic groups with ethyl acetate fraction of *T. grandis* reduced the elevated blood glucose levels significantly (*p* < 0.05) compared with the DBC.

#### 2.4.3. Effect on Oral Glucose Tolerance Test

The oral glucose tolerance test (OGTT) is used to evaluate glucose intolerance and is presented in Figure 5. Glucose tolerance was significantly (*p* < 0.05) impaired in the DBC group. The treatment with ethyl acetate fraction of *T. grandis* glucose significantly (*p* < 0.05) improved the glucose tolerance in the diabetic rats.

#### 2.4.4. Effect on Liver Glycogen and Carbohydrate Metabolism Enzymes

The liver glycogen content was significantly (*p* < 0.05) depleted in the DBC group but upon treatment, the glycogen content of the liver significantly (*p* < 0.05) increased in diabetic-treated groups (DTGL, DTGH, and DBM). Both α-amylase and liver glucose-6-phosphatase activity were significantly increased in the DBC group (*p* < 0.05). The treatment with plant fraction significantly reduced the activity of these enzymes (Table 2).

#### 2.4.5. Effect on Indices of Hepatic and Renal Damages

The serum level of creatine, urea, uric acid, AST, ALT, and ALP levels was significantly elevated (*p* < 0.05) in the DBC group compared to the NC group, as depicted in Table 2. However, treating the *T. grandis* significantly reduced the level in diabetic rats. No significant differences were observed between the NC and NTPI groups for all these parameters.

#### 2.4.6. Effect on Diabetes Indexes (Serum Insulin, Insulin Resistance, and Pancreatic β-Cell Function)

The induction of diabetes by partial destruction of pancreatic β cells resulted in a significant (*p* < 0.05) decrease in serum insulin concentration and HOMA-β scores, as observed in the DBC group (Table 2). Upon treatment, the serum insulin concentration and HOMA-β scores were increased significantly (*p* < 0.05), as observed in DTGL, DTGH, and DBM rats. Likewise, HOMA-IR scores and the serum fructosamine concentration were significantly lower (*p* < 0.05) in treated rats (DTGL, DTGH, and DBM) when compared with untreated rats (DBC). Still, they were significantly (*p* < 0.05) higher than the normal rats (NC and NTTG).

#### 2.4.7. Effect on Pancreatic β-Cell Morphology

The histopathological analysis of the pancreas is represented in Figure 6. The pancreas pancreatic β-cell morphology of DBC showed a shrinkage of the pancreatic β islets with a concomitant reduction in the β cells, as compared to the pancreatic β-cell morphology of NC pancreas that showed the normal and distinct appearance of the pancreatic islets with the significant number of β cells, which looked lightly stained than the surrounding acinar cells. Treatment of the diabetic group (DTGL, DTGH, and DBM) showed improved morphology and regeneration of pancreatic islets and β cells. However, DTGH showed the best morphology in this regard.

### 2.5. Phytochemical Constituents

GCMS analysis led to the identification of compounds shown in Table 3 and Figure 7.

## 3. Discussion

Over the years, herbal medicine has gained tremendous attention in global healthcare delivery [17]. Accumulative evidence suggests that polyphenols are essential in preventing and managing T2D via two approaches: (1) The insulin-dependent approaches; this involves the pancreatic islet β-cell protection from hyperglycemia-induced oxidative stress [18,19], promotion of β-cell proliferation [20], activation of insulin signalling [21], and stimulation of pancreatic insulin secretion [22]. (2) Insulin-independent approaches include inhibition of glucose absorption [23], inhibition of digestive enzymes [24,25], increased disposal of postprandial glucose [26], and regulation of intestinal microbiota [27]. Therefore, the possibilities of new plant-based drugs for diabetes are not far-fetched. This study exploits the phytochemical composition of different extracts of *T. grandis* leaves viz-a-viz their antidiabetic activity and determines the antidiabetic activity (insulin and non-insulin independent parameters) of the extracts with the highest activity.

Inhibition of carbohydrate-digesting enzymes is one of the mechanisms that plant polyphenols use to modulate glycaemic activity. A previous study attributed the inhibition of α-glucosidase by methanol extract of *T. grandis* flowers to its polyphenolic constituents [28]. The ethanol crude extract, demonstrating the highest inhibitory activity, was chosen for further partitioning in different solvents for further analysis.

The extract from different fractions of the ethanol crude extract followed the same pattern as the crude extract regarding the polyphenols and the consequent inhibitory activity of α-glucosidase (Table 1 and Figure 2). This further corroborates the role of polyphenols in modulating α-glucosidase. Once dietary carbohydrate is digested into glucose, it will be absorbed into the bloodstream. Delay in glucose absorption is a therapeutic approach to controlling postprandial hyperglycaemia [29,30]. It has been reported that plant polyphenols inhibit glucose absorption at the lumen of the small intestine [23,31]. The ability of different extracts from different fractions of ethanol crude extract to delay glucose absorption in intestinal jejunum in this study depends on their polyphenol extracts in these fractions (Figure 2A,B). Our observation was in accordance with the recent research that showed significant inhibitory activities against α-glucosidase in different fractions of *T. grandis* [13].

Furthermore, 70% of postprandial glucose insulin-mediated disposal occurs in peripheral tissue such as skeletal muscle [32,33]. However, the role of polyphenols in postprandial glucose insulin-mediated disposal is contradictory. Some studies suggested that polyphenols suppressed glucose uptake [34,35,36], while some claimed that they enhanced glucose uptake [13,30,37]. The present study supported that the polyphenols have a relationship with the glucose uptake in the peripheral tissue (Figure 2C). To further exploit the other mechanism of antidiabetic activity of *T. grandis*, the ethyl acetate extract (because it demonstrated the highest activity out of all the fractions) was selected for studies in fructose–streptozotocin (STZ)-induced diabetic rats.

Polydipsia, polyuria, and polyphagia with progressive decreasing body weight are some of the overt symptoms of diabetes [38], which were also observed in the diabetic groups of our experiment (Figure 3A,B), indicating a successful induction of diabetes in this study. Similar studies have linked these parameters to prolonged and stable T2D conditions [39,40]. A similar trend was observed in the present study, where the treatment with *T. grandis* not only increased the weight loss (Figure 3B) but also ameliorated polydipsia and polyphagia to a non-diabetic state (Figure 3A).

Persistent hyperglycaemia is a clinical manifestation of T2D. Reports have linked elevated blood glucose to insulin insufficiency, which could result from pancreatic β-cell mass reduction and peripheral tissues and cells’ inability to respond to insulin action [41,42,43]. It was observed in our study that untreated diabetic rats showed elevated blood glucose (Figure 4) with low-serum insulin (Table 2) and sullen pancreatic morphology (Figure 6). The administration of *T. grandis* reduced the elevated blood glucose (Figure 4) with corresponding improvements in β-cell functions (HOMA-IR) and a rise in serum insulin (Table 1). In addition, the ability of *T. grandis* to improve pancreatic β-cell morphology (Figure 6) indicates its anti-hyperglycaemia potential. T2D is characterised by β-cell dysfunction and insulin resistance. The insensitivity of peripheral cells characterises glucose intolerance to insulin action [44,45]. The administration of *T. grandis* in treated diabetic groups improved glucose tolerance and clearance (Figure 5), as evident in the low HOMA-IR of the treated diabetic groups (Table 2).

The liver plays a crucial role in glucose homeostasis, accounting for about 25% of glucose disposal [46]. In normal physiological conditions, the liver responds to insulin by taking up elevated glucose and stores it as glycogen. In diabetic conditions, the excess glucose is diverted to de novo fatty acid synthesis, which could result in hepatic steatosis [47]. A previous study reported a reduction in the liver glycogen level as a pathological feature of the induction of diabetes in experimental rats [48]. The increased liver glycogen concentration (Table 2) further supports the improved glucose tolerance potential of *T. grandis* (Figure 5).

In diabetic subjects, hepatic glucose production is increased in the postabsorptive state and fails to be adequately regulated by insulin due to excessive gluconeogenesis [49]. Previous studies showed that insulin inhibits hepatic glucose production [50,51,52], which gives further credence to our present study, as a high blood glucose level in diabetic rats (Figure 4) corresponds to the low-serum insulin (Table 2). Glucose-6-phosphatase is an enzyme and rate-limiting step in gluconeogenesis [53]. An increase in the activity of glucose-6-phosphatase signals gluconeogenesis rather than glycolysis, as seen in untreated diabetic rats (Table 2). The ability of *T. grandis* to decrease the activity of glucose-6-phosphatase significantly (Table 2) demonstrates decreased gluconeogenesis in diabetic rats, further supporting the improved pancreatic morphologic (Figure 6), increased β-cell functions and serum insulin concentration (Table 2), as is evident in the deceased blood glucose (Figure 4).

Prolonged hyperglycaemia has been linked to diabetic complications through oxidative stress, involving a cascade of steps, such as the hexosamine pathway, advanced glycation end product (AGE), polyol pathway, and activation of protein kinase C [54,55]. One of the products of glycated protein, an early stage of AGE, is fructosamine [56], the increase in blood glucose level (Figure 4) and a corresponding increase in the serum fructosamine level, as seen in the untreated diabetic rats (Table 2), which may suggest the onset of glycation cascade reaction and diabetic complications. The administration of *T. grandis* reduced the fructosamine level in the diabetic-treated groups, further suggesting the ameliorative potential of *T. grandis* towards glycation cascade and diabetic complications.

In a diabetic state, amino acids from protein catabolism flow are transported to the liver to increase gluconeogenesis and accelerate ureagenesis, resulting in hypoproteinemia and hypoalbuminemia [57,58]. Consequently, the serum creatinine, urine total protein, and urine albumin levels are elevated. Also, increases in uric acid and urea serum levels have been reported to be responsible for the development of nephropathy [59]. The increased serum level of uric acid and urea in untreated diabetic rats was ameliorated by treating with *T. grandis* (Table 2), further supporting the ability of *T. grandis* to reduce the progression of diabetic complications.

The above biological activities of *T. grandis* could be attributed to the phytochemical constituents of the plant (Table 3 and Figure 7). A previous study reported the inhibition of α-glucosidase and α-amylase by methanol extracts of *T. grandis* flowers and attributed it to its polyphenolic [14]. The present study identified Stigmasterol and its derivates in the leaves of ethyl acetate and ethanol extracts (Figure 7), which have been reported to inhibit α-glucosidase and α-amylase [60,61]. Interestingly, the synergistic action of β-sitosterol and stigmasterol [62] in streptozotocin-induced diabetic rats has been reported. Likewise, 9,12-Octadecadienoic acid, ethyl ester has also been reported as a constituent of *Mahonia leschenaultia* Takeda [63], delta Decalactone in *Zygophyllum album* [64], which all exert antihyperglycemic and antihyperlipidemic effects in streptozotocin-induced diabetic rats.

## 4. Material and Methods 

### 4.1. Plant Material

The leaves of *T. grandis* were collected around March 2016 at Owo community in Ondo State, Nigeria. The Voucher specimen number PSB 174 was assigned after authentication by the Department of Plant Science and Biotechnology, Adekunle Ajasin University, Akungba, Nigeria. The plant’s name was checked with http://www.theplantlist.org (http://www.theplantlist.org/tpl1.1/search?q=Tectona+grandis+, accessed on 24 June 2023). The plant leaves were washed to remove dirt and air-dried to constant weight. The dried samples were pulverized into fine powder and stored in air-tight containers for further analysis.

### 4.2. General Experimental Procedures

The general experimental procedure is represented as shown in Figure 8. Briefly, the dried leaves samples were defatted using hexane and sequentially extracted with the following solvent according to their polarity (ethyl acetate → ethanol → distilled water). After that, an in vitro antidiabetic assay was performed on the individual extract to determine the extract with the highest activity. The extract with the highest activity was chosen for further fractionate with solvents of increasing polarity (dichloromethane, ethyl acetate, ethanol, butanol, and aqueous). The fractions obtained were then screened regarding their antidiabetic activity in vitro, intestinal glucose absorption, and muscle glucose uptake in ex vivo experiments. Also, the fraction with the highest activity was then investigated for the possible mechanism of action in fructose–streptozotocin (STZ)-induced diabetic rats.

### 4.3. Sequential Extraction of Crude Extract

A dried leaves sample of *T. grandis* was pounded into a fine powder and defatted with hexane. Thereafter, it was sequentially extracted in ethyl acetate, ethanol, and water. For each solvent extraction, the sample was soaked overnight, and for 48 h at room temperature with gentle agitation, the extract was filtered and concentrated at 40 °C under reduced pressure using a rotary evaporator (Buchi Rotary Evaporator, Buchi Corporation, Flawil, Switzerland). The extract was air-dried to a fine powder and stored at −20 °C. 

### 4.4. Fractionation of the Crude Extract

The extract was reconstituted in water/methanol (1:9 ratio) and sequentially partitioned with dichloromethane, ethyl acetate, and butanol to yield their respective fractions. The remaining residue was collected as an aqueous fraction. The fractions were concentrated to dryness using a vacuum rotary evaporator while the aqueous fraction was concentrated in a water bath at 40 °C and stored at 4 °C for subsequent analysis.

#### 4.4.1. Estimation of Total Phenol Content 

Total phenol content was determined by the method described by Antolovich et al. [65]. Briefly, 200 μL of various crude extracts/fractions (240 μg/mL concentration) and garlic acid of various concentrations was incubated with 1 mL of 10-times-diluted Folin Ciocalteau reagent and 800 μL of 0.7 M Na_2_CO_3_ at room temperature for 30 min. The absorbance was measured at 765 nm and the total phenolic extract extrapolated from the gallic acid standard curve as mg/g gallic acid equivalent (GAE)

#### 4.4.2. Estimation of α-Glucosidase Inhibitory Activity

The α-glucosidase enzyme inhibitory activity of the crude extracts/factions was measured according to a previously established protocol [66] with slight modifications. A 250 μL aliquot of different concentrations (15–240 μg/mL) of the extracts/fractions or acarbose was incubated for 10 min with 500 μL of 1.0 U/mL Yeast α-glucosidase solution in 100 mM phosphate buffer (pH 6.8) at 37 °C. Then, 250 μL of 5 mM pNPG solution in phosphate buffer (100 mM, pH 6.8) was added, and the mixture was further incubated at 37 °C for 20 min. After that, the absorbance was measured at 405 nm, and enzyme inhibition was calculated as per the following formula:$$\%inhibitory\;activity=\frac{Absorbance\;of\;control-Absorbance\;of\;sample}{Absorbance\;of\;control}\times 100$$

### 4.5. Ex Vivo Antidiabetic Assay

Twelve-week-old male Sprague-Dawley rats were procured from the Biomedical Research Unit of the University of Kwazulu-Natal, Westville Campus, Durban, South Africa. The animals were euthanized by decapitation, and the psoas muscle and the whole gastrointestinal tract (GIT) were harvested for use in glucose uptake and glucose absorption, respectively. All animal protocols were in accordance with the guidelines of the Animal Research Ethics Committee of the University of KwaZulu-Natal, Durban 4000, South Africa (ethical approval number: AREC/067/017D)

#### 4.5.1. Estimation of Intestinal Glucose Absorption

The effect of the crude extracts/fractions on intestinal glucose absorption was determined according to a previously described protocol [67]. The jejunal segments of the harvested small intestine were cut into smaller pieces of 5 cm, and the inner jejunal lumen was inverted and cleaned with 2 mL of Kreb’s buffer (118 mM NaCl, 5 mM KCl, 1.328 mM CaCl_2_·2H_2_O, 1.2 mM KH_2_PO_4_, 1.2 mM MgSO_4,_ and 25 mM NaHCO_3_) via an antiseptic syringe. Thereafter, each piece was incubated with different concentrations of the crude extracts/fractions, 8 mL of Kreb’s buffer containing 11.1 mM of glucose for 2 h at 5% CO_2_, 95% oxygen, and 37 °C in a Steri-Cult CO_2_ incubator (Labotec, Midrand, South Africa). Then, 1 mL of the mixture was taken before and after the 2 h incubation period, and the amount of glucose in the mixture was measured using Automated Chemistry Analyzer (Labmax Plenno, Lago Santa, Brazil). The intestinal glucose absorption was calculated using the following formula:Abdominal glucose absorption = GC1 − GC2/length of jejunum (cm)

#### 4.5.2. Estimation of Muscle Glucose Absorption

The effect of the crude extracts/fractions on muscle glucose uptake was determined according to the previously described protocol [67]. A 0.5 g freshly collected psoas muscle was incubated in 8 mL of Kreb’s buffer solution containing the increasing concentration of fractions (30, 60, 120, and 240 μg/mL) and 11.1 mM of glucose, with or without insulin (50 mU/mL), in a CO_2_ incubator under 5% CO_2_ and 95% oxygen at 37 °C for 1 h. Then, 2 mL of aliquot was collected from each incubating sample to determine the glucose concentration. The muscle glucose uptake was calculated as per the following formula:Muscle glucose uptake= (GC1 − GC2)/0.5 g of muscle tissue
GC1 = glucose concentrations before incubation
GC2 = glucose concentrations after incubation

### 4.6. In Vivo Antidiabetic Assay

#### 4.6.1. Experimental Animals

Sprague-Dawley (SD) rats (forty-two) of six-weeks-old, with an average weight of 193 g, were obtained and housed at the Biomedical Resource Unit, University of Kwazulu-Natal, Durban, South Africa, and were kept following the regulations of the Animal Ethics Regulation Committee (AREC) of the University of Kwazulu-Natal, Durban, South Africa (Ethical approval No AREC/067/017D).

#### 4.6.2. Animal Grouping and Induction of Diabetes

After one week of acclimatization, the rats were randomly selected and split into six groups of seven animals, as illustrated below.

Normal control group (NC);Normal toxicological control + 300 mg/Kg BW of the fraction dissolved in 5% DMSO (NTTG);Diabetic control group (DBC);Diabetic + low dose (150 mg/kg BW dissolved in 5% DMSO) of the fraction (DTGL);Diabetic + high dose (300 mg/kg BW dissolved in 5% DMSO) of the fraction (DTGH);Diabetic + metformin (300 mg/kg BW dissolved in 5% DMSO) (DBM).

The animals in diabetic groups (DBC, DTGL, DTGH, and DBM) were administered 10% fructose solution ad libitum for two weeks to mimic insulin resistance. This was followed by the administration of a single injection of streptozotocin (STZ) (40 mg/kg) BW in citrate buffer (pH 4.5) intraperitoneally to cause partial destruction of pancreatic β cells according to the method developed by Wilson and Islam, 2012. The animals in normal groups (NC and NTAI) were administered normal water and citrate buffer instead of 10% fructose and STZ, respectively. After a week, all the animals’ non-fasting blood glucose (NFBG) was measured from the tail vein using a portable glucometer (Glucoplus Inc. Canada). Animals with NFBG greater than 11.1 mmol/L were considered diabetic, while those with NFBG less than 11.1 mmol/L were excluded from the diabetic groups.

#### 4.6.3. Intervention Period

The diabetic rats were administered their respective dose daily (as shown in the animal grouping) using an oral gastric gavage needle. At the same time, rats in the control group were administered a similar volume of the vehicle (5% DMSO). During this period, daily intake of food and fluid was measured. Also, the body weight and NFGB were observed weekly in all the groups, and the intervention period lasted for four weeks.

#### 4.6.4. Oral Glucose Tolerance Test (OGTT)

The oral glucose tolerance test was carried out immediately after the intervention period. After overnight fasting (12 h), the rats were given an oral dose of glucose solution (2 g/kg BW). The blood glucose levels were observed at 0 (just before oral glucose administration), 30, 90, and 120 min (after the oral glucose administration) using a portable glucometer (Glucoplus Inc., Saint-Laurent, QC, Canada). The area under the curve (AUC) was calculated according to the following formula.
$${\mathrm{AUC}}_{t_{k}}=\sum_{i=1}^{k}\left(\frac{C_{i-1}+C_{i}}{2}\right)\left(t_{i}-t_{i-1}\right)$$
where, *C_i_* = the blood glucose concentration at the time *t_i_*.

#### 4.6.5. Collection of Blood, Serum, and Organs

The rats in all the groups were sacrificed by anaesthesia using isofor during the experiment. The blood was collected from each animal through cardiac puncture into a plain tube and centrifuged for 15 min at 300 rpm to obtain the serum, which was preserved at −20 °C for further analysis. The liver and pancreas of each rat were obtained, washed, weighed, and stored at −20 °C for further analysis. A little piece of the pancreatic tissue of each animal was preserved for histopathological examination in 10% formalin solution.

#### 4.6.6. Biochemical Analysis

The ultrasensitive rat insulin enzyme-linked immunosorbent assay (ELISA) kit (Mercodia, Upsala, Sweden) following the kit manual was used to measure the serum insulin concentration, while automated Chemistry Analyzer (Labmax Plenno, Labtest Co. Ltd., Lagoa Santa, Brazil) was used to measure the serum lipid profile (total cholesterol, HDL and LDL cholesterol, and triglycerides), fructosamine, urea, and creatinine concentrations, as well as liver function enzymes: aspartate and alanine aminotransferases (AST and ALT) and alkaline phosphatase (ALP).

#### 4.6.7. Liver Glycogen Estimation

Liver glycogen was estimated according to the method of Roehrig and Allred [68] with slight modifications. Briefly, 0.3 g of the liver tissue was cut and digested with 0.5 mL of 30% potassium hydroxide saturated with sodium sulphate (Na_2_SO_4_) after boiling for 30 min. The resulting solution was immediately cooled on ice. Thereafter, 670 µL of 95% ethanol was allowed to stand on ice for 30 min to precipitate the glycogen and centrifuged at 840 g for 30 min. The supernatant was discarded, and the residue was then re-suspended in 300 µL of 95% ethanol and centrifuged for 20 min to precipitate the glycogen further. The glycogen precipitate obtained was dissolved in 1 mL of distilled water. An aliquot of 20 µL was taken and made up to 200 µL with distilled water in a tube. Thereafter, 200 µL of 5% phenol was added to the aliquot and the glycogen standards (10, 20, 30, 40, 50, 60, 70, 80, 90 µg/mL) followed by 1 mL of sulphuric acid (96–98%). They were boiled in a water bath for 10 min. The tubes were allowed to cool for 10 min, and absorbance was read at 490 nm in a plate reader (Synergy HTX Multi-mode reader, Bio Tek Instrument Inc., Winooski, VT, USA). The glycogen content was calculated as µg/mg from the glycogen standard curve.

#### 4.6.8. Determination of Liver Glucose-6-Phosphatase Activity

The glucose-6-phosphatase activity was determined by spectrophotometric measurement of inorganic phosphate production using the method described by Baginski et al. [69] with slight modification. Briefly, 300 µL of 0.5 M citrate buffer (pH 6.5) and 100 µL of 0.2 M glucose-6-phosphate solution were mixed in two separate microtubes, marked as test and blank mixtures, and incubated at 37 °C for 5 min. Thereafter, 100 µL of the liver homogenate was added to the test mixture. The test and the blank mixtures were, respectively, incubated for 5 min at 37 °C, followed by the addition of 1000 µL of 10% trichloroacetic acid (TCA). They were chilled on ice for 10 min and centrifuged at 300 rpm for 10 min to obtain the supernatant. The aliquot was taken from the supernatant, and the absorbance was read at 340 nm in a plate reader (Synergy HTX Multi-mode reader, Bio Tek Instrument Inc., Winooski, VT, USA). The glucose-6-phosphatase activity was expressed as units/min/mg of protein tissue.

#### 4.6.9. Homeostatic Model Assessment

Fasting blood glucose concentration (FGB) and serum insulin concentrations were used for homeostatic model assessment scores. The score measures the pancreatic β-cell function (HOMA- β) and the level of insulin resistance (HOMA-IR). The calculation of HOMA- β and HOMA-IR is given below:HOMA-IR = [Fasting serum insulin (U/L) × Fasting blood glucose (mmol/L)]/22.5
HOMA-β = [20 × Fasting serum insulin (U/L)]/Fasting blood glucose (mmol/L) − 3.5 1 U/L = 7.174 pmol/L for insulin concentration.

#### 4.6.10. Histological Examination of Pancreatic Tissue

Further, 4 µm of the tissue sections was cut and fixed on the slide. The slides were then deparaffinized and rinsed in water. Slides were stained in haematoxylin for 5 min and rinsed with water, and they were counter-stained in eosin, mounted in DPX, coverslipped, and viewed with a Leica slide scanner (SCN 4000, Leica Biosystems, Wetzlar, Germany).

### 4.7. Gas Chromatography–Mass Spectrometric (GC–MS) Analysis

The ethyl acetate fraction of *T. grandis* ethanol extract of the leaves was subjected to GC–MS analysis. The GC–MS analysis was conducted with an Agilent Technologies 6890 Series GC and (an Agilent, Santa Clara, CA, USA) 5973 Mass Selective Detector. An HP-5 MS capillary column (30 m × 0.25 mm ID, 0.25 μm film thickness, 5% phenylmethylsiloxane) was used. A flow rate of 1.0 mL/min and 37 cm/sec linear velocity of an ultra-pure helium gas as carrier gas were used. The injector temperature was set at 250 °C with an initial oven temperature of 60 °C set to 280 °C at a rate of 10 °C/min, with a hold time of 3 min. Injections of 1 μL were made in splitless mode with a split ratio 20:1.

The mass spectrometer was operated in electron ionization mode at 70 eV and electron multiplier voltage at 1859 V. Other MS operating parameters were as follows: ion source temperature was 230 °C; quadrupole temperature was 150 °C; solvent delay was 4 min; and scan range was 50–70 amu. Compounds were identified by direct comparison of the retention times and mass spectral data with those in the NIST library.

### 4.8. Statistical Analysis

Data were represented as means ± standard deviation of 5 animals and analysed with GraphPad prism version 8 using Tukey’s HSD multiple range post hoc test. *p* < 0.05 was considered as significant.

## 5. Conclusions

This study compared the antidiabetic potential of different solvent extractions of *T. grandis* and fractionated the crude extracts that demonstrated the best activity through solvent–solvent partitioning. The ethyl acetate fraction of the ethanol crude extract showed the best antidiabetic activity in inhibiting α-glucosidase, delaying glucose absorption at the small intestine’s lumen and enhancing the muscle’s postprandial glucose uptake. Also, the in vivo studies of the ethyl acetate fraction of the ethanol crude extract in diabetic rats indicated the ability of *T. grandis* to improve pancreatic histology, insulin secretion, and sensitivity, thus reducing postprandial hyperglycaemia within the safety used at a dose of 300 mg/kg.bw. These activities could be attributed to the phytochemical constituents of the plant. Future studies will examine, in detail, the molecular mechanism(s) and metabolic pathways that may be involved in the anti-hyperglycaemia activity of *T. grandis.*

## Figures and Tables

**Figure 1 ijms-25-00028-f001:**
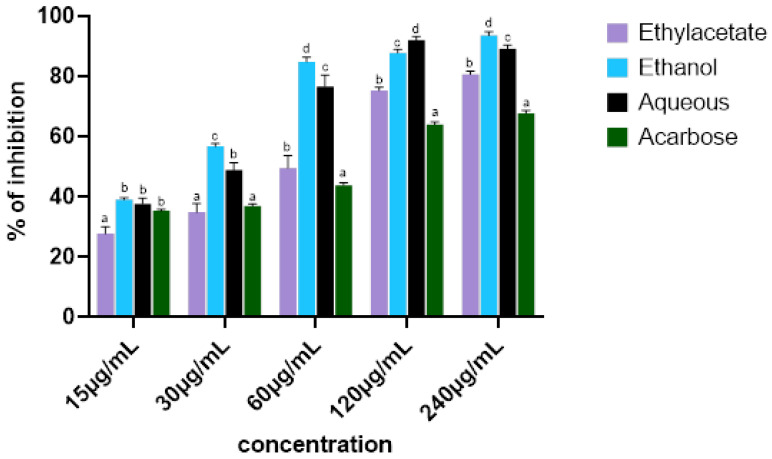
α-glucosidase enzyme inhibitory activity of crude extracts of different parts of *T. grandis.* Values represent mean ± standard deviation (*n* = 3). ^a–d^ Different alphabets over the bars for a given concentration represent significance differences (*p* < 0.05).

**Figure 2 ijms-25-00028-f002:**
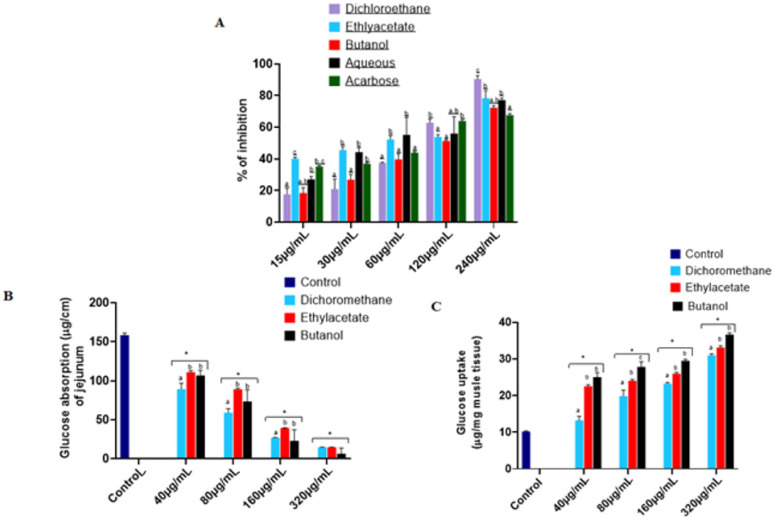
Antidiabetic effects of different solvent fractions of *T. grandis* ethanol leaves extract on (**A**) α-glucosidase enzyme inhibitory activity, (**B**) glucose absorption in isolated rat jejunum (**C**) glucose uptake in psoas rat muscle. Values represent mean ± standard deviation (*n* = 3). ^a–c^ Different alphabets over the bars for a given concentration for each extract represent significance differences (*p* < 0.05) * Significantly different from control (*p* < 0.05).

**Figure 3 ijms-25-00028-f003:**
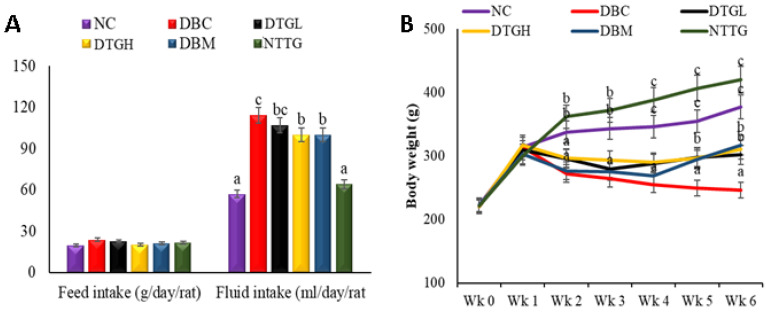
Effect of oral treatment of ethyl acetate fraction of *T. grandis* on (**A**) body weight (**B**) Food and fluid intake. Data are presented as the mean ± SD of 5 animals. ^a–c^ Different alphabets over the bars for a given parameter represent represent significance differences (*p* < 0.05). NC, Normal Control; DBC, Diabetic Control; DTGL, Diabetic *T. grandis* low dose; DTGH, Diabetic *T. grandis* high dose; DBM, Diabetic Metformin; NTTG, Normal *T. grandis* high dose.

**Figure 4 ijms-25-00028-f004:**
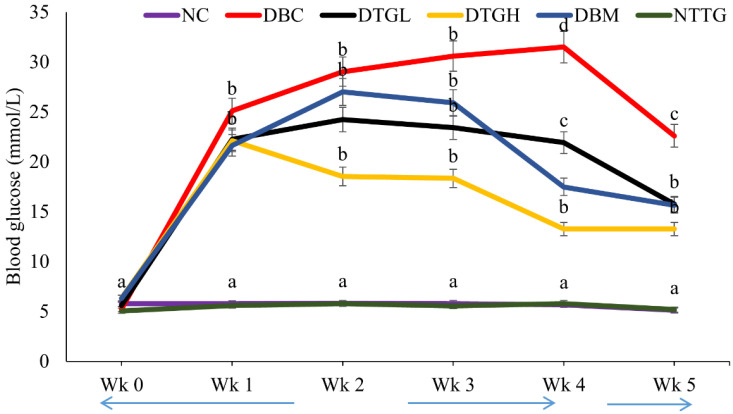
Effect of oral treatment of ethyl acetate fraction of *T. grandis* on weekly blood glucose in different animal groups. Data are presented as the mean ± SD of 5 animals. ^a–d^ Different alphabets near the lines for a given time represent significance differences (*p* < 0.05). NC, Normal Control; DBC, Diabetic Control; DTGL, Diabetic *T. grandis* low dose; DTGH, Diabetic *T. grandis* high dose; DBM, Diabetic Metformin; NTTG, Normal *T. grandis* high dose NFGB, non-fasting blood glucose.

**Figure 5 ijms-25-00028-f005:**
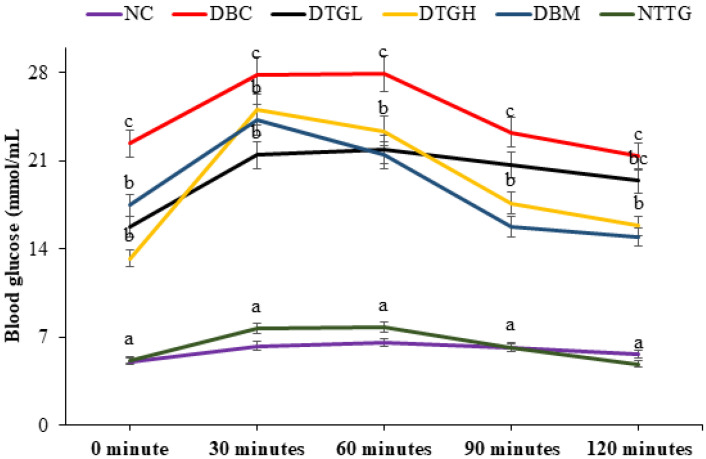
Effect of oral treatment of ethyl acetate fraction of *T. grandis* on oral glucose tolerance test in different animal groups during the experimental period. Data are presented as the mean ± SD of 5 animals. ^a–c^ Different alphabets near the lines for a given time represent significance differences (*p* < 0.05). NC, Normal Control; DBC, Diabetic Control; DTGL, Diabetic *T. grandis* low dose; DTGH, Diabetic *T. grandis* high dose; DBM, Diabetic Metformin; NTTG, Normal *T. grandis* high dose.

**Figure 6 ijms-25-00028-f006:**
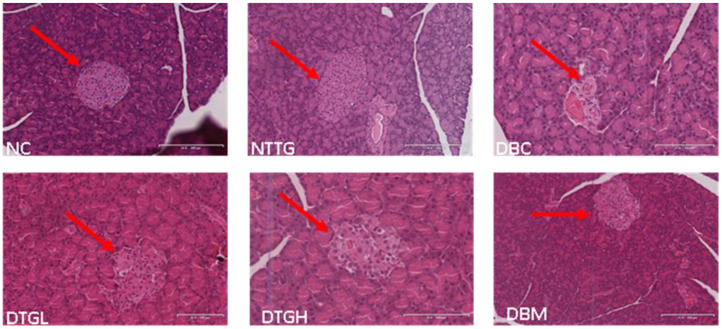
Effect of oral treatment of ethyl acetate fraction of *T. grandis* on histological examinations (20×, 100 µm) of the pancreas in different animal groups during the experimental period. The NC and NTTG groups have the highest number of β-cells per islet. The red arrow points to the islet of Langerhans. The islet of the DBC group is distorted with the fewest number of β-cells. DTGL, DTGH, and DBM groups showed regeneration of the islet with more β-cells than DBC. NC, Normal Control; DBC, Diabetic Control; DTGL, Diabetic *T. grandis* low dose; DTGH, Diabetic *T. grandis* high dose; DBM, Diabetic Metformin; NTTG, Normal *T. grandis* high dose.

**Figure 7 ijms-25-00028-f007:**
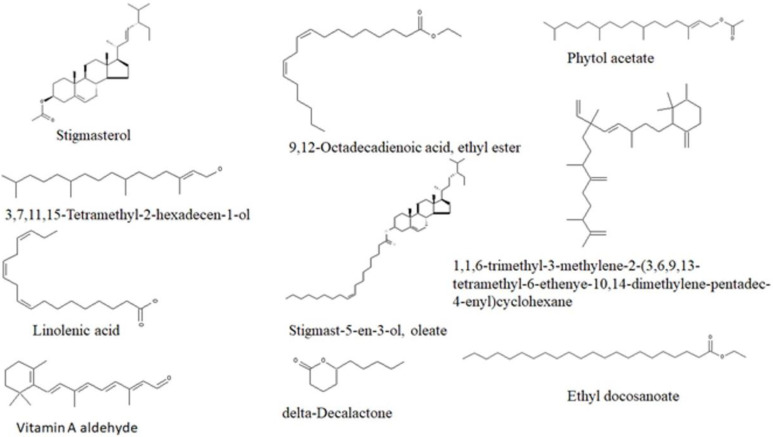
Identified compounds in ethyl acetate fraction of *T. grandis* by GC-MS.

**Figure 8 ijms-25-00028-f008:**
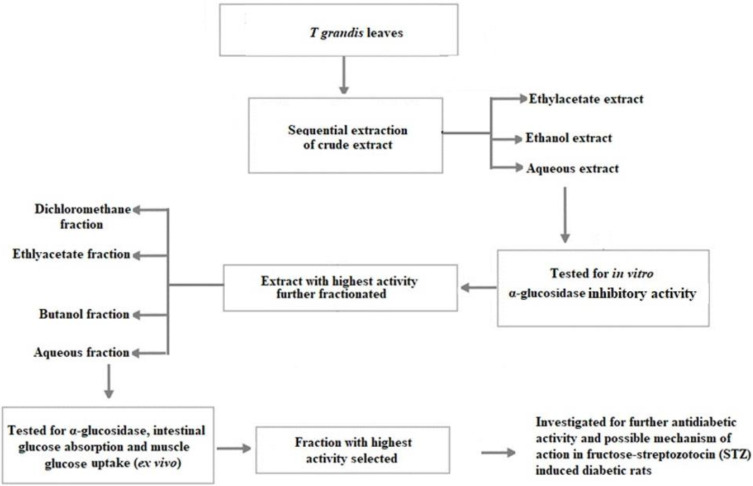
General experimental design.

**Table 1 ijms-25-00028-t001:** IC_50_ values of different extracts/fraction and total phenol content.

Crude Extract	α-Glucosidase IC_50_ Values (µg/mL)	Total Phenol Content (mg/g GAE)
Ethyl Acetate	32.17 ± 4.46 ^b^	10.34 ± 3.87 ^b^
Ethanol	16.34 ± 5.12 ^a^	25.84 ± 6.53 ^a^
Aqueous	28.43 ± 6.05 ^b^	14.75 ± 2.94 ^c^
**Fractions extract**		
Dichloromethane	67.61 ± 3.42 ^c^	38.72 ± 7.03 ^d^
Ethyl acetate	36.31 ± 2.12 ^a^	132.55 ± 6.25 ^a^
Butanol	91.20 ± 2.14 ^d^	51.91 ± 4.13 ^c^
Aqueous	51.29 ± 3.05 ^b^	73.25 ± 5.97 ^b^

Data represent mean ± standard deviation of triplicate determinations. ^a–d^ Different alphabets within a column for a given parameter are significantly different from each other (Tukey’s-HSD multiple range post hoc test, *p* < 0.05) IC_50_ = Concentration to inhibit 50% activity; GAE = garlic acid equivalent.

**Table 2 ijms-25-00028-t002:** Biochemical parameters in fructose–streptozotocin diabetic rats studies.

	NC	DBC	DTGL	DTGH	DBM	NTTG
Insulin (ρmol/L)	72.60 ± 2.47 ^d^	31.61 ± 3.20 ^a^	49.99 ± 3.29 ^b^	61.81 ± 4.35 ^c^	59.23 ± 1.70 ^c^	74.81 ± 0.72 ^d^
HOMA-IR	1.85 ± 0.76 ^a^	8.61 ± 1.54 ^c^	5.95 ± 1.1 ^b^	3.06 ± 0.46 ^b^	4.32 ± 0.96 ^b^	1.92 ± 0.54 ^a^
HOMA-β	64.34 ± 2.45 ^d^	2.9 ± 0.32 ^a^	32.01 ± 1.96 ^c^	18.93 ± 3.67 ^b^	13.59 ± 2.34 ^b^	67.11 ± 1.23 ^d^
Liver glycogen (mg/g tissue)	5.36 ± 0.31 ^e^	1.63 ± 0.88 ^a^	2.32 ± 0.54 ^b^	4.18 ± 0.26 ^d^	3.58 ± 0.18 ^c^	5.04 ± 0.66 ^e^
Glucose-6-phosphatase (U)	1.92 ± 0.03 ^a^	6.21 ± 0.08 ^e^	2.51 ± 0.22 ^d^	2.21 ± 0.03 ^c^	2.01 ± 0.03 ^ab^	1.98 ± 0.04 ^a^
Fructosamine F (µmol/L)	119 ± 3.31 ^a^	867 ± 11.24 ^c^	534.13 ± 14 ^b^	144 ± 15.34 ^b^	633 ± 17.28 ^b^	114 ± 6.88 ^a^
Urea (mg/dL)	54.75 ± 5.21 ^a^	75.00 ± 3.12 ^c^	61.60 ± 2.27 ^b^	59.45 ± 3.04 ^b^	61.25 ± 1.25 ^b^	55.75 ± 4.48 ^a^
Uric acid (mg/dL)	1.14 ± 0.17 ^a^	4.80 ± 0.92 ^c^	2.35 ± 0.54 ^b^	2.53 ± 0.21 ^b^	1.95 ± 0.14 ^b^	0.75 ± 0.53 ^a^
Creatine (U/L)	2.13 ± 0.03 ^a^	2.81 ± 0.08 ^d^	2.41 ± 0.22 ^bc^	2.3 ± 0.03 ^b^	2.61 ± 0.03 ^b^	2.23 ± 0.04 ^b^

Results are presented as mean ± SD of 5 rats. ^a–e^ Values with different alphabets along a row are significantly different from each other (Tukey’s-HSD multiple range post hoc test, *p* < 0.05). NC, Normal Control; DBC, Diabetic Control; DTGL, Diabetic *T. grandis* low dose; DTGH, Diabetic *T. grandis* high dose; DBM, Diabetic Metformin; NTTG, Normal *T. grandis* high dose. HOMA-β = (Fasting serum insulin in U/l × 20/Fasting blood glucose in mmol/L − 3.5); HOMA-IR = [(Fasting serum insulin in U/l × Fasting blood glucose in mmol/L)/22.5]; Glucose-6-phosphatase 1 U = 1 μmole phosphate liberated per min per mg of the liver tissue.

**Table 3 ijms-25-00028-t003:** Identified compounds in ethyl acetate and ethanol crude extract of *T. grandis* by GC–MS.

Retention Time (min)	Compounds	Relative Abundance (%)
**Ethyl Acetate**		
14.39	Phytol, acetate	2.59
15.14	2,6,10,15-Tetramethylheptadecane	1.19
15.57	2-Ethyl-2-methyl-1-tridecanol	1.81
17.09	Phytol	5.17
19.00	2-methylhexacosane	2.03
20.75	Urs-12-en-28-oic acid, 3-hydroxy-, methyl ester, (3.beta.)-	1.97
24.12	Ursane-3,12-diol	1.05
24.23	Cholane-5,20(22)-diene-3b-phenoxy	3.46
24.31	22-Stigmasten-3-one	6.82
**Ethanol**		
4.06	Linolenic acid	3.24
14.83	Phytol, acetate	2.68
15.12	3,7,11,15-Tetramethyl-2-hexadecen-1-ol	2.4
15.96	delta-Decalactone	8.07
16.55	Hexadecanoic acid, ethyl ester	2.71
17.59	9,12-Octadecadienoic acid, ethyl ester	12.21
17.82	Linolenic acid	7.17
19.42	1,1,6-trimethyl-3-methylene-2-(3,6,9,13-tetramethyl-6-ethenye-10,14-dimethylene-pentadec-4-enyl)cyclohexane	0.38
20.19	Ethyl arachidate	2.67
21.23	Ethyl docosanoate	2.69
24	Stigmast-5-en-3-ol, oleate	1.15
24.39	Stigmasterol acetate	1.15
24.73	Vitamin A aldehyde	0.52

## Data Availability

Data is contained within the article.

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
