# Peer review of "Ethyl Acetate Fractions of Tectona Grandis Crude Extract Modulate Glucose Absorption and Uptake as Well as Antihyperglycemic Potential in Fructose–Streptozotocin-Induced Diabetic Rats"

_ijms, 2023, doi:10.3390/ijms25010028_

Round 1

Reviewer 1 Report

Comments and Suggestions for Authors

In this study the Authors address the antihyperglycemic effects of ethyl acetate fractions of Tectona grandis crude extract in fructose-streptozotocin-induced diabetic rats.

The investigation of the effects of “in vivo” fractions of T. grandis crude extract treatment, underlying its role in Type 2 diabetes, is interesting to discover alternative therapeutic strategies against diabetes and related diseases.

The authors show that fractions of Tectona grandis crude extract inhibit α-glucosidase, decreasing glucose absorption at the small intestine's lumen and enhancing the muscle's postprandial glucose uptake in “ex vivo” experiments. Furthermore, ethyl acetate fraction treatment significantly reduces high blood glucose levels in diabetic rats stimulating insulin secretions and improving insulin sensitivity, incresing liver glycogen content, decreasing α-amilase and glucose 6 phosphatase activity and ameliorating many biochemical parameters.

Unfortunately, however, the weak feature of the manuscript concerns the failure to provide the molecular mechanisms involved; the authors should at least to study some crucial kinases involved in glucose metabolism such as AMPK and Akt and analyze the expression levels of enzymes such as glucosidase, enzymes related to glycogen metabolism, glucose 6 phosphatase and GLUT4.

Moreover, there are several pitfalls disseminated throughout the text, with a poor English style and incorrect typing (signed in yellow).

Based on these drawbacks, I believe the manuscript requires major revisions to reach the threshold for publication on International Molecular Sciences.

Other minor points:

25 lines, improved is reapet three times

39-40-41 lines: β-ells

43 lines: add refs

Uniform T. grandis (T, grandis; T grandis)

Uniform temperaturse as 40°C

Fig 1 and formula have a very low definition

124 lines kerb buffer

134 lines 11.1mM with space

Abbreviation for body weight is BW

Fig 3c muscle not musle

188 lines: fructosamine not frutosamine

262 lines: the sentence starts with but and there is no punctuation

333 lines: the sentence starts with but

380 lines: change dotting

399 lines: two times different

Comments on the Quality of English Language

Moderate editing of English language required

Reviewer 2 Report

Comments and Suggestions for Authors

The article by Sanni et al. described the anti-diabetic potential of a Tectona grandis ethyl acetate extract by using in vitro, ex vivo and in vivo models. Although the topic is of some interest, there are too many methodological problems that mean that the work, in my opinion, cannot be published in International Journal of Molecular Sciences

Major concerns

1)     Treatment of rats with high dose of streptozotocin is widely used to induce both insulin-dependent and noninsulin-dependent diabetes mellitus presently by inducing pancreatic β-cell death. This rat model mimic type 1 diabetes and not type 2 diabetes. There are some animal models mimicking type 2 diabetes that should be used in the place of the model utilized by the authors

2)     All the discussion is based on the anti-diabetic potential of phenolic compounds; however the authors did not demonstrate the presence of these compounds in the extract. GC-MS analysis is not enough to chemically characterize the extract. The authors should perform LC-MS analysis

3)     The authors should motivate the choice to use only glucosidase inhibitory assay for the screening of the extracts. In my opinion is not a correct selection procedure.

4)     Moreover, the calculated IC50 values for crude extracts are quite strange. The aqueous crude extract displayed an inhibitory activity against glucosidase near to 80% at 60 μg/mL (Figure 2) but the calculated IC50 value was about 110 μg/mL (?)(Table 1). The same consideration for the ethyl acetate extract (with about 50% of inhibition at 30 μg/mL (Figure 2) and the calculated IC50 value was about 135 μg/mL (Table 1)) and ethanol extract that overcame the 50% of inhibition already at 30 μg/mL (Figure 2) but the reported IC50 value was about 56 μg/mL (Table 1). This poses serious problems regarding the quality of the selection

Minor point

Line 105 page 5: Which type of glucosidase was used by the authors? The behaviour is very different between yeast and mammalian glucosidase.

Line 110 page 5: “Scavenging activity” should be “Inhibitory activity”

Line 124 page 6: 11.1 mmol/L of glucose and not mL.  How the authors set the glucose concentration for the absorption and transport assays?

Line 167 page 7: What was the vehicle? Water:methanol (1:9) solution? This should be clearly stated in the manuscript

Line 266 page 11: inhibited should be increased

The manuscript should be double checked to eliminate the language mistake

Comments on the Quality of English Language

The manuscript should be double checked to eliminate the language mistake

Reviewer 3 Report

Comments and Suggestions for Authors

The authors are encouraged to rewrite the manuscript:

- following the editor guideline
- improving english or seek the help of english speaking colleagues
- figures are difficult to read to check the article content
- material and methods are insufficiently described
- the authors should emphasize the originality of their work when comparing to scientific litterature that already described their findings
Best regards
The reviewer

Please see attached for additional comments.

Comments on the Quality of English Language

improve english or seek the help of english speaking colleagues

Round 2

Reviewer 1 Report

Comments and Suggestions for Authors

The authors have improved the paper and now it can be published on IJMS

Best regards

Reviewer 2 Report

Comments and Suggestions for Authors

The authors answered to most of the comments; however there are still some opened questions which make the article unsuitable for publication.

1)     The authors did not justify their choice to use only α-glucosidase assay as screening for the crude fractions. It is clear that ethanol extract was a better inhibitor of α-glucosidase, however, it is not possible to exclude a priori that the aqueous extract or the ethyl acetate extract are more effective in inhibiting the absorption of glucose or in enhancing the transport of glucose. The screening of the crude extracts should be done by considering all these aspects.

2)     The use of yeast glucosidase is not relevant from a physiological point-of-view. The inhibition of yeast and mammalian α-glucosidases is very different and specific to the type of substrate. To be relevant, the inhibitory assay should be carried out with mammalian α-glucosidase.

3)     I can’t find in the manuscript the data about the total phenolic content of the extracts neither the methods used to quantify the total phenolic content in materials and methods. In my opinion it makes no sense to base the discussion on the antidiabetic effect of phenolic compounds (without demonstrating their presence) and then show identification data on other non-phenolic compounds. Are phenolic compounds present in the tested ethyl acetate extract? If they are present they are non-volatile compounds and have to be identify by LC-MS.

Comments on the Quality of English Language

The English quality of the manuscript has been improved although there are still some typos

Reviewer 3 Report

Comments and Suggestions for Authors

Dear authors
The reviewer suggest that you reconsider the advices provided on the first review. The manuscript adjustments are not sufficient including the reference section that still does not comply with IJMS guidline for authors.
Best Regards

Comments on the Quality of English Language

Moderate editing of English language required

Round 3

Reviewer 2 Report

Comments and Suggestions for Authors

no comments

Reviewer 3 Report

Comments and Suggestions for Authors

Dear Authors,
Some minor improvement were performed in the texte but the reference section is still not following the appropriate IJMS author guideline.

Best regards

Comments on the Quality of English Language

Minor editing required